EZH2/H3K27Me3 and phosphorylated EZH2 predict chemotherapy response and prognosis in ovarian cancer

Sun Si 1
Yang Qiang 1
Cai E 1
Huang Bangxing 2
Ying Feiquan 1
Wen Yiping 1
Cai Jing 1
Yang Ping yangping5127@163.com 1 3
1 Department of Gynecology and Obstetrics, Union Hospital, Tongji Medical College, Huazhong University of Science and Technology , Wuhan , China
2 Department of Pathology, Union Hospital, Tongji Medical College, Huazhong University of Science and Technology , Wuhan , China
3 Department of Obstetrics and Gynecology, First Affiliated Hospital, School of Medicine, Shihezi University , Shihezi , China
Kurita Takeshi
Electronic publication date: 2020 May 12
Publication date: 2020
Volume: 8
Electronic Location ID: e9052
Received 2019 Dec 5; Accepted 2020 Apr 3
Copyright: ©2020 Sun et al.
Copyright year: 2020
Copyright holder: Sun et al.
License: This is an open access article distributed under the terms of the Creative Commons Attribution License, which permits unrestricted use, distribution, reproduction and adaptation in any medium and for any purpose provided that it is properly attributed. For attribution, the original author(s), title, publication source (PeerJ) and either DOI or URL of the article must be cited.
License URL: https://creativecommons.org/licenses/by/4.0/

Keywords: Ovarian cancer, Prognosis, Chemotherapy response, Phosphorylated EZH2, EZH2, H3K27me3, P-Akt1, Overall survival, Progression-free survival, HGSOC

Funding: National Natural Science Foundation of China 81572572 81702570 81702575 Scientific and Technological Research Projects of Xinjiang Production and Construction Corp 2017DB012 Union Hospital Scientific Research Fund 2018-229 This work was supported by the National Natural Science Foundation of China (81572572, 81702570 and 81702575), the Scientific and Technological Research Projects of Xinjiang Production and Construction Corp (2017DB012) and the Union Hospital Scientific Research Fund (2018-229). The funders had no role in study design, data collection and analysis, decision to publish, or preparation of the manuscript.

==============================
Background

EZH2 acts as an oncogene through canonical pathway EZH2/H3K27Me3 and uncanonical pathway pAkt1/pS21EZH2 in many solid tumors including ovarian cancer. However, the clinical value of EZH2/H3K27Me3 and pAkt1/pS21EZH2 remain unclear. In the current study, we aim to investigate the correlation between these two pathways to clinical-pathological parameters and prognosis.

Methods

EZH2, H3K27Me3, pAkt1 and pS21EZH2 expression were evaluated by tissue micro-array and immunohistochemistry in a cohort of ovarian cancer patients. The results were analyzed based on clinical characteristics and survival outcomes.

Results

EZH2, H3K27Me3, pAkt1 and pS21EZH2 were universally expressed in ovarian cancer specimens with a positive expression rate of 81.54% (53/65), 88.89% (48/54), 63.07% (41/65) and 75.38% (49/65). EZH2-pS21EZH2 (Spearman r = 0.580, P < 0.0001) and pS21EZH2-pAkt1 (Spearman r = 0.546, P < 0.0001) were closely correlated while EZH2- H3K27Me3 were less closely correlated (Spearman r = 0.307, P = 0.002). Low pS21EZH2 associated with better chemotherapy response (OR = 0.184; 95% CI [0.052–0.647], P = 0.008) according to logistic regression with an area under the curve of 0.789 (specificity 89.36%, sensitivity 68.42%) by ROC analysis and predicted improved progression-free survival (HR = 0.453; 95% CI [0.229–0.895], P = 0.023) as indicated by multivariate cox regression. A combination of EZH2low/H3K27Me3low status predicted better chemotherapy response (OR = 0.110; 95% CI [0.013–0.906], P = 0.040) and better progression-free survival (HR = 0.388; 95% CI [0.164–0.917], P = 0.031). The results suggested that EZH2/H3K27Me3 and pEZH2 predicted chemotherapy response and progression-free survival in ovarian cancer.

Introduction

Ovarian cancer is the most lethal gynecological malignancy (Torre et al., 2018). Cytoreductive surgery combined with platinum based chemotherapy remained primary regimen for ovarian cancer. Resistance to platinum-based chemotherapy was the main cause that led to chemotherapy failure (Christie & Bowtell, 2017). Promotion of drug research and development aimed to improve the prognosis of ovarian cancer patients was fruitful. Combination of topotecan and sorafenib significantly increased the progression-free survival in women with platinum-resistant ovarian cancer (Chekerov et al., 2018). In combination of carboplatin and paclitaxel, nintedanib increased progression-free survival for advanced ovarian cancer patients (Du Bois et al., 2016). Despite the encouraging achievement of the new drugs, the overall survival of ovarian cancer patients was not significantly improved. Customization of appropriate chemotherapy regimen was based on pre-chemotherapy sensitivity prediction. Therefore, pre-chemotherapy prediction for platinum resistance is essential in stratification of patients to different primary chemotherapy in order to avoid treatment delay and strive for more benefit.

As the key component of polycomb repressive complex 2 (PRC2), enhancer of zeste homolog 2 (EZH2) contributes to epithelial malignancies through histone modification and epigenetic gene silencing. EZH2 was intimately involved in platinum resistance according to previous studies: EZH2 was overexpressed in cancer stem-like cells enriched by platinum (Wen et al., 2017); EZH2-H3K27Me3 axis induced chromatin condensation, SLFN11 gene silencing, DNA-damage repair deficiency and acquired platinum resistance (Gardner et al., 2017). Generally, EZH2 overexpression was associated with platinum resistance and poor prognosis of epithelial malignancies (Hu et al., 2010; Sun et al., 2018; Yi et al., 2017). However, whether non-canonical EZH2 associated tumorigenic pathway was involved in platinum resistance was still not clear.

Non-canonical tumorigenic mechanisms of EZH2 included STAT3/Akt1/pS21EZH2 mediated transcriptional activation and AMPK/pT311EZH2 mediated attenuation of PRC2 dependent H3K27Me3 (Chen et al., 2016; Wan et al., 2018). Previous pathological immune-histochemical analysis reported that higher pT311EZH2 correlated with favorable survival in ovarian cancer patients (Wan et al., 2018). Yet, the clinical implication of pS21EZH2 remained unknown. To further understand the role of canonical and non-canonical EZH2 associated pathways in chemotherapy resistance and prognosis, we investigated the correlation between the key components of two EZH2 pathways EZH2-H3K27Me3 and pAkt1-pS21EZH2 independently and in combination to the clinical outcomes of ovarian cancer patients.

Materials and Methods

Ethics statement

The study protocol was approved by the Ethical Committee of Union Hospital, Tongji Medical College, Huazhong University of Science and Technology (IOGR No: IORG0003571).

Clinical-pathological features

Non-stratified, non-matched clinical and pathological data were retrospectively collected. Basic characteristics included patient age, histology, International Federation of Gynecology and Obstetrics (FIGO) stage, tumor type, treatment regimen, chemotherapy response and follow-up data. Chemotherapy resistance was defined by relapse within six months after completing chemotherapy or progression during the primary chemotherapy. Relapses were diagnosed on clinical symptoms, radiological evidence and biochemical abnormalities such as elevated CA125. Overall survival (OS) was defined as the time from date of diagnosis to death or last follow-up date, progression-free survival (PFS) was defined as the time from surgery to relapse or last follow-up date. Diagnoses of all patients were confirmed pathologically.

Patient and tumor characteristics

Tissue samples from a total of 65 patients were obtained from ovarian cancer patients admitted to Union Hospital, Tongji Medical College, Huazhong University of Science and Technology between August 2008 and October 2015. All patients underwent radical surgery followed by standard platinum based chemotherapy. The median age of the patients was 52 (32–72) years.

Tissue micro-array construction

65 non-consecutive, unselected primary ovarian cancer specimens were included in the tissue microarray. The tumor samples were collected within one hour after resection from the primary site. Formalin-fixed paraffin-embedded (FFPE) tissue blocks were prepared according to the standard procedure. Tissue cylinders of 2 mm in diameter were punched from representative areas of each block with regard to the matching H&E staining control by a MiniCore Control Station (Alphelys Sarl, France). The Selected tissue cylinders were re-arranged and brought into three paraffin blocks by a semi-automated tissue arrayer (Beecher Instruments, Sun Prairie, WI, USA). 4 µm section slides were prepared for further use.

Immunohistochemistry

Immunohistochemistry (IHC) was performed as previously described. Briefly, the slides were dewaxed in xylene and went through a serial of descending ethanol to rehydrate. Antigen retrieval was performed through microwave irradiation. Blocking and staining were performed using Histostain Kits (SP9001 and SP9002, ZSGB-Bio, Beijing, China). Primary antibodies for EZH2 (1:100, Cell Signaling Technology #5246, Danvers, USA), pS21EZH2 (1:100, Bethyal #00388, Montgomery, USA), p S473Akt1 (1:100, Cell Signaling Technology #4060, Danvers, USA) and H3K27Me3 (1:400, Abclonal #A2363, Boston, USA) and were applied as recommended by the manufacturers. Replacements of primary antibodies by IgG were provided as negative and isotype controls. DAB color development and hematoxylin counterstaining were performed as appropriate.

IHC evaluation

IHC evaluation was performed by two trained researchers in a blinded manner, a third pathologist was referred to when disagreement occurred. IHC scores (0–12) were assessed using a semi-quantitative scale multiplying the percentage of positive tumor cells (0, 0%; 1, <25%; 2, 25%–50%; 3, 50%–75%; 4, >75%) by stain intensity (0, negative; 1, weak; 2, moderate; 3, strong). Only nuclear staining was considered valid when measuring the scores for EZH2, H3K27Me3, p-EZH2 and p-Akt1. All results were based on five high power (400X) fields.

Statistical analysis

All statistical analyses were performed using SPSS 20.0 software. The association between the IHC scores of EZH2, H3K27Me3, p-Akt1 and p-EZH2 and ovarian cancer clinical pathological characteristics were assessed using chi square, Fisher’s exact and Kruskal-Wallis tests. The correlation of EZH2/H3K27Me3 and p-EZH2/p-Akt1 were analyzed using Spearman test. Mann Whitney test was used to compare the difference between two groups as appropriate. Receiver operating characteristic (ROC) curve were plotted to examine the value of EZH2, p-EZH2 and EZH2/H3K27Me3 combination as predictive factor for chemotherapy response. Logistic and cox regression were used to analyze risks for chemotherapy response and survival. For measurement of chemotherapy response, odds ratio (OR) was used. When OR >1, the patients were prone to chemo-resistant; when OR <1, the patients were prone to chemo-sensitive. For measurement of OS and PFS, hazard ratio (HR) was used. When HR >1, the patients were prone to worse outcome; When HR <1, the patients were prone to better outcome. Kaplan–Meier method and log-rank test were used to plot and analyze survival curves. P values <0.05 were considered statistically significant.

Results

Association between EZH2 associated pathways and clinical pathological features

To investigate the association of canonical and non-canonical EZH2 pathways to clinical pathological features, we first investigated each pathway component independently. The components of canonical and non-canonical EZH2 pathways were universally expressed in ovarian cancer specimens with a positive expression rate of 81.54% (53/65) for EZH2, 88.89% (48/54) for H3K27Me3, 63.07% (41/65) for pAkt1, and 75.38% (49/65) for pS21EZH2. EZH2-pS21EZH2 (Spearman r = 0.580, P < 0.0001) and pS21EZH2-pAkt1 (Spearman r = 0.546, P < 0.0001) were closely correlated while EZH2-H3K27Me3 were less closely correlated (Spearman r = 0.307, P = 0.002) (Fig. 1). Samples were classified as EZH2 High/Low, H3K27Me3 High/Low, pEZH2 High/Low and pAkt1 High/Low groups by the median IHC scores (6 for EZH2, 8.5 for H3K27Me3, 6 for pS21EZH2 and 2 for p-Akt1). High EZH2 (P = 0.053) and pEZH2 (P = 0.011) expression were closely related to chemotherapy resistance (Fig. 2). Specimens from FIGO stage I-II patients were prone to higher pAkt1 level (P = 0.097). H3K27Me3 IHC score tended to be lower in tissue samples from elderly aged patients (P = 0.057). No significant association was found between EZH2, H3K27Me3, pS21EZH2 or pAkt1 and histology or tumor type (Table 1).

Figure 1 General expression of EZH2/H3K27Me3 and pAkt1/pEZH2 in ovarian cancer.

(A–D) Representative images of EZH2, H3K27Me3, pAkt1 and pEZH2 staining in ovarian cancer tissue (mainly stained in nucleus). 40 × and 200 × (E) and (F) Positive correlation between the IHC score of pAkt1 with pEZH2 and EZH2 with H3K27Me3 (Spearman’s correlation test).

Figure 2 pEZH2 predicted chemotherapy response in ovarian cancer.

(A) The corresponding histogram of semi-quantification of pEZH2 level in resistant and sensitive groups. Mann Whitney test, P < 0.001. (B) ROC curves of EZH2, pEZH2, the combination of EZH2low/H3K27Me3low and the combination of pAkt1low/pEZH2low. AUCEZH2 = 0.688, AUCpEZH2 = 0.789, AUC EZH2low/H3K27Me3low = 0.6461, AUCpAkt1low/pEZH2low = 0.6341.

Table 1 Association between pEZH2, pAkt1, EZH2 and H3K27Me3 to clinical pathological features in ovarian cancer by IHC-score stratification.a

Clinicopathological Features	N	pEZH2	pAkt1	EHZ2	N	H3K27Me3	
		High	Low	P	High	Low	P	High	Low	P		High	Low	P	
Age (years)b															
<50	27	13	14	0.883	16	11	0.213	15	12	1.000	23	16	7	0.057	
≥50	38	19	19		16	22		20	18		31	13	18		
Histology															
Serous	53	24	29	0.215	23	30	0.747	28	25	0.761	45	24	21	0.598	
Others	12	8	4		4	8		7	5		9	5	4		
FIGO stage															
I–II	17	7	10	0.574	13	5	0.097	7	10	0.266	14	7	7	0.766	
III–IV	48	25	23		23	27		28	20		40	22	18		
Tumor type															
Type I	15	9	7	0.574	10	6	0.250	9	7	1.000	12	7	5	0.755	
Type II	50	23	26		21	28		26	23		42	22	20		
Chemo response															
Sensitive	46	18	28	0.011	27	19	0.404	20	26	0.053	38	19	19	0.363	
Resistant	18	14	4		8	10		13	5		15	10	5		
Notes.

a The median IHC score was chosen as the cut-offs for pEZH2, pAkt1 and H3K27Me3.

b Age at surgery.

P values were calculated by chi square and Fishers exact tests.

In order to assess the clinical relevance of different EZH2 related pathways, we next analyzed clinical pathological features in tissues with different EZH2/H3K27Me3 and pAkt1/pS21EZH2 levels. For EZH2-H3K27Me3, samples were classified as EZH2high/H3K27Me3high, EZH2high/H3K27Me3low, EZH2low/H3K27Me3high and EZH2low/H3K27Me3low. For pAkt1-pS21EZH2, samples were classified as pAkt1high/pEZH2high, pAkt1high/pEZH2low, pAkt1low/pEZH2high and pAkt1low/pEZH2low. The distribution of number of cases in each group were compared in relation to clinical features including age, histology, FIGO stage, tumor type and chemo-response. The results suggested that EZH2/H3K27Me3 level (P = 0.053) and pAkt1/pEZH2 level (P = 0.025) were closely correlated with chemotherapy response (Tables 2 and 3).

Table 2 Association between EZH2/H3K27Me3 to clinical pathological features in ovarian cancer by IHC-score stratification.a

Clinicopathological Features	EZH2high/H3K27Me3high	EZH2high/H3K27Me3low	EZH2low/H3K27Me3high	EZH2low/H3K27Me3low	P	
	N	(100%)	N	(100%)	N	(100%)	N	(100%)		
Age (years)b	16	100	14	100	13	100	11	100	0.312	
<50	9	56.25	4	28.57	7	53.85	3	27.27		
≥50	7	23.75	10	71.43	6	46.15	8	72.73		
Histology	16	100	14	100	13	100	11	100	0.447	
Serous	12	75.00	12	85.71	12	92.31	9	81.82		
Others	4	25.00	2	14.29	1	7.96	2	18.18		
FIGO stage	16	100	14	100	13	100	11	100	0.537	
I–II	3	18.75	4	28.57	4	30.77	3	27.27		
III–IV	13	81.25	10	71.43	9	69.23	8	72.73		
Tumor type	16	100	14	100	13	100	11	100	0.682	
Type I	5	31.25	2	14.29	2	15.38	3	27.27		
Type II	11	68.75	12	85.71	11	84.62	8	72.73		
Chemo response	16	100	13	100	13	100	11	100	0.057	
Sensitive	10	62.50	8	61.54	4	30.77	11	100		
Resistant	6	37.50	5	38.46	9	69.23	0	0		
Notes.

a The median IHC score was chosen as the cut-offs for pEZH2, pAkt1 and H3K27Me3.

b Age at surgery.

P values were calculated by Kruskal–Wallis test.

Table 3 Association between pAkt1/pEZH2 to clinical pathological features in ovarian cancer by IHC-score stratification.a

Clinicopathological Features	pAkt1high/pEZH2high	pAkt1high/pEZH2low	pAkt1low/pEZH2high	pAkt1low/pEZH2low	P	
	N	(100%)	N	(100%)	N	(100%)	N	(100%)		
Age (years)b	18	100.00	9	100.00	14	100.00	24	100.00	0.899	
<50	6	33.33	5	55.56	7	50.00	9	37.50		
≥50	12	66.67	4	44.44	7	50.00	15	62.50		
Histology	18	100.00	9	100.00	14	100.00	24	100.00	0.907	
Serous	15	83.33	8	88.89	10	71.43	20	83.33		
Others	3	16.67	1	11.11	4	28.57	4	16.67		
FIGO stage	18	100.00	9	100.00	14	100.00	24	100.00	0.250	
I–II	2	11.11	3	33.33	5	35.71	7	29.17		
III–IV	16	88.89	6	66.67	9	64.29	17	70.83		
Tumor type	18	100.00	9	100.00	14	100.00	24	100.00	0.999	
Type I	5	27.78	1	11.11	4	28.57	6	25.00		
Type II	13	72.22	8	88.89	10	71.43	18	75.00		
Chemo response	18	100.00	9	100.00	14	100.00	23	100.00	0.025	
Sensitive	11	61.11	8	88.89	7	50.00	20	86.96		
Resistant	7	38.89	1	11.11	7	50.00	3	13.04		
Notes.

a The median IHC score was chosen as the cut-offs for pEZH2, pAkt1 and H3K27Me3.

b Age at surgery.

P values were calculated by Kruskal-Wallis test.

EZH2/H3K27Me3 and pS21EZH2 predicted platinum-based chemotherapy response

As both EZH2/H3K27Me3 and pAkt1/pS21EZH2 pathways correlated with chemotherapy response, subsequent assessment of the prognostic value of these two pathways were performed. First, the prognostic value of independent EZH2, H3K27Me3, pS21EZH2 and pAkt1 for chemotherapy response was assessed by logistic regression and ROC analysis. Logistic regression revealed that patients with lower pS21EZH2 (OR = 0.184; 95%CI [0.052–0.647], P = 0.008) and lower EZH2 (OR = 0.095; 95%CI [0.948–10.107], P = 0.061) IHC scores had better chemotherapy response (Table 4). The area under the curve (AUC) for EZH2 and pS21EHZ2 were 0.688 (specificity 74.47%, sensitivity 63.16%) and 0.789 (specificity 89.36%, sensitivity 68.42%) (Fig. 2). Although H3K27Me3 and pAkt1 did not have significant prognostic value as single factors, combined with EZH2 and pS21EZH2, EZH2low/H3K27Me3low (OR = 0.110; 95%CI [0.013–0.906].107, P = 0.040) and pAkt1low/pEZH2low (OR = 0.260; 95%CI [0.066–1.023], P = 0.054) status predicted better chemotherapy response (Table 4). For patients with serous ovarian cancer, pS21EZH2 remained prognostic for chemotherapy response (OR = 0.188; 95%CI [0.050–0.705], P = 0.013). Patients with EZH2low/H3K27Me3low (OR = 0.133; 95%CI [0.016–1.133], P = 0.065) and pAkt1low/pEZH2low (OR = 0.288; 95%CI [0.070–1.190], P = 0.086) were prone to better chemotherapy response (Table 5).

Table 4 Associations of different parameters and chemotherapy response.

	Chemotherapy response	
	OR	95% CI	P	
Age				
≥ 50	1			
<50	1.224	0.413–3.747	0.697	
Histology				
Non-serous	1			
Serous	2.222	0.436–11.324	0.337	
FIGO stage				
III–IV	1			
I–II	0.258	0.052–1.272	0.096	
Tumor Type				
Type II	1			
Type I	0.810	0.223–2.945	0.748	
EZH2 expression				
High	1			
Low	0.323	0.099–1.055	0.061	
pEZH2 expression				
High	1			
Low	0.184	0.052–0.647	0.008	
H3K27Me3 expression				
High	1			
Low	0.500	0.144–1.741	0.276	
pAkt1 expression				
High	1			
Low	0.880	0.293–2.641	0.819	
EZH2/H3K27Me3 expression				
Others	1			
Low EZH2/Low H3K27Me3	0.110	0.013–0.906	0.040	
pAkt1/pEZH2 expression				
Others	1			
Low pAkt1/Low pEZH2	0.260	0.066–1.023	0.054	

Table 5 Associations of different parameters and chemotherapy response in serous ovarian cancer.

	Chemotherapy response	
	OR	95% CI	P	
Age				
≥50	1			
<50	1.089	0.331–3.577	0.888	
FIGO stage				
I–II	1			
III–IV	2.692	0.516–14.038	0.240	
Tumor Type				
Type I	1			
Type II	2.429	0.311–18.986	0.398	
EZH2 expression				
High	1			
Low	0.364	0.105–1.263	0.111	
pEZH2 expression				
High	1			
Low	0.188	0.050–0.705	0.013	
H3K27Me3 expression				
High	1			
Low	0.417	0.106–1.644	0.211	
pAkt1 expression				
High	1			
Low	1.029	0.314–3.369	0.963	
EZH2/H3K27Me3 expression				
Others	1			
Low EZH2/Low H3K27Me3	0.133	0.016–1.133	0.065	
pAkt1/pEZH2 expression				
Others	1			
Low pAkt1/Low pEZH2	0.288	0.070–1.190	0.086	

EZH2/H3K27Me3 and pS21EZH2 predicted PFS in ovarian cancer patients

Since chemotherapy response was an independent factor influencing prognosis, we next evaluate whether EZH2 associated pathways representing varied chemotherapy response status were of prognostic significance to survival of ovarian cancer patients. In all the factors included for analysis, Advanced FIGO stage and poor chemotherapy response were closely related with poorer OS and PFS through both uni- and multi-variate analyses (Tables 6 and 7). Kaplan–Meier survival analysis and univariate cox regression analysis for OS revealed that lower EZH2 (HR = 0.464; 95%CI [0.205–1.048], P = 0.065) and pS21EZH2 (HR = 0.464; 95%CI [0.206–1.046], P = 0.064) staining tended to correlate with better OS. A combination of EZH2low/H3K27Me3low staining was significantly associated with improved OS (HR = 0.257; 95%CI [0.076–0.863], P = 0.028). Kaplan–Meier survival analysis and univariate cox regression analysis for PFS revealed that lower pS21EZH2 staining was significantly associated with better PFS (HR = 0.477; 95%CI [0.226–0.882], P = 0.020). Similarly, a combination of EZH2low/H3K27Me3low staining was significantly associated with increased PFS (HR = 0.344; 95%CI [0.145–0.813], P = 0.015) (Fig. 3). When adjusted for FIGO stage, multivariate Cox regression analysis for PFS revealed that pS21EZH2 staining (HR = 0.453; 95%CI [0.229–0.895], P = 0.023) and EZH2low/H3K27Me3low status (HR = 0.388; 95%CI [0.164–0.917], P = 0.031) remained to be independent prognostic factors for PFS. For patients with serous ovarian cancer, chemotherapy response was closely related with OS and PFS through univariate analyses. Lower pS21EZH2 staining (HR = 0.431; 95%CI [0.207–0.896], P = 0.024) and pAkt1low/pEZH2lowstatus (HR = 0.479; 95%CI [0.219–1.046], P = 0.065) were associated with improved PFS (Table 8). We did not found significant relation of age, histology, tumor type, H3K27Me3 expression and pAkt1 expression to OS or PFS through uni- or multi-variate analyses.

Table 6 Univariate Cox regression survival analysis.

	OS	PFS	
	HR	95% CI	P	HR	95% CI	P	
Age							
≥50	1			1			
<50	1.224	0.573–2.701	0.581	1.205	0.624-2.328	0.579	
Histology							
Non-serous	1			1			
Serous	1.346	0.463–3.909	0.585	1.735	0.672–4.481	0.255	
FIGO stage							
III–IV	1			1			
I–II	0.118	0.016–0.870	0.036	0.371	0.144–0.955	0.040	
Tumor Type							
Type II	1			1			
Type I	0.790	0.316–1.975	0.615	0.757	0.343–1.668	0.489	
Chemo-response							
Resistant	1			1			
Sensitive	0.088	0.035–0.219	0.000	0.046	0.018–0.114	0.000	
EZH2 expression							
High	1			1			
Low	0.464	0.205–1.048	0.065	0.586	0.300–1.146	0.118	
pEZH2 expression							
High	1			1			
Low	0.464	0.206–1.046	0.064	0.447	0.226–0.882	0.020	
H3K27Me3 expression							
High	1			1			
Low	0.823	0.342–1.980	0.664	0.707	0.345–1.445	0.341	
pAkt1 expression							
High	1			1			
Low	0.985	0.443–2.192	0.971	1.008	0.518–1.963	0.980	
EZH2/H3K27Me3							
Others	1			1			
EZH2low/H3K27Me3low	0.257	0.076–0.863	0.028	0.344	0.145–0.813	0.015	
pAkt1/pEZH2							
Others	1			1			
pAkt1low/pEZH2low	0.561	0.224–1.407	0.218	0.523	0.252–1.086	0.082	

Table 7 Multivariate Hazard Cox regression survival analysis.

	OS	PFS	
	HR	95% CI	P	HR	95% CI	P	
EZH2 expression							
High	1			1			
Low	0.535	0.237–1.209	0.133	0.649	0.331–1.272	0.208	
pEZH2 expression							
High	1			1			
Low	0.512	0.227–1.155	0.107	0.453	0.229–0.895	0.023	
H3K27Me3 expression							
High	1			1			
Low	1.029	0.415–2.555	0.950	0.734	0.358–1.505	0.399	
pAkt1 expression							
High	1			1			
Low	1.157	0.524-2.555	0.718	1.115	0.571-2.176	0.750	
EZH2/H3K27Me3							
Others	1			1			
EZH2low/H3K27Me3low	0.333	0.099-1.121	0.076	0.388	0.164-0.917	0.031	
pAkt1/pEZH2							
Others	1			1			
pAkt1low/pEZH2low	0.547	0.219–1.369	0.198	0.504	0.243–1.049	0.067	

Figure 3 pEZH2 and EZH2/H3K27Me3 predicted prognosis in ovarian cancer.

(A–F) Kaplan-Meier plots for overall survival and progression-free survival in ovarian cancer patients with different pEZH2 (A and B), EZH2 (C and D) and EZH2/H3K27Me3 (E and F) level.

Table 8 Univariate Cox regression survival analysis of serous ovarian cancer patients.

	OS	PFS	
	HR	95% CI	P	HR	95% CI	P	
Age							
≥50	1			1			
<50	1.231	0.531–2.853	0.628	1.134	0.558–2.304	0.728	
FIGO stage							
III–IV	1			1			
I–II	0.238	0.032–1.802	0.165	0.544	0.189–1.570	0.260	
Tumor Type							
Type II	1			1			
Type I	0.949	0.217–4.145	0.945	1.391	0.420–4.609	0.589	
Chemo-response							
Resistant	1			1			
Sensitive	0.084	0.029–0.237	0.000	0.051	0.019–0.138	0.000	
EZH2 expression							
High	1			1			
Low	0.627	0.266–1.480	0.287	0.791	0.389–1.609	0.518	
pEZH2 expression							
High	1			1			
Low	0.482	0.201–1.155	0.102	0.431	0.207–0.896	0.024	
H3K27Me3 expression							
High	1			1			
Low	0.734	0.266–2.031	0.552	0.645	0.295–1.410	0.272	
pAkt1 expression							
High	1			1			
Low	0.866	0.364–2.059	0.744	0.923	0.455–1.876	0.826	
EZH2/H3K27Me3							
Others	1			1			
EZH2low/H3K27Me3low	0.379	0.111–1.292	0.121	0.486	0.203–1.163	0.105	
pAkt1/pEZH2							
Others	1			1			
pAkt1low/pEZH2low	0.498	0.182–1.357	0.173	0.479	0.219–1.046	0.065	

Discussion

In the present study, we investigated the prognostic value of canonical pathway EZH2/H3K27me3 and non-canonical pathway pAkt1/pS21EZH2 of EZH2 in ovarian cancer. We first assessed the association of independent EZH2, H3K27me3, pAkt1, pS21EZH2 and EZH2/H3K27me3, pAkt1/pS21EZH2 combination with clinical characteristics of ovarian cancer patients such as age, histology, FIGO stage, tumor type and chemotherapy response. Then the correlation of the key components of two EZH2 pathways EZH2/H3K27Me3 and pAkt1/pS21EZH2 to chemotherapy response, OS and PFS of ovarian cancer patients were assessed independently and in combination. We observed that EZH2low/H3K27Me3low predicted better chemotherapy response, OS and PFS while low pS21EZH2 predicted poor chemotherapy response and PFS. The results suggested that both canonical and non-canonical EZH2 pathways contributed to chemotherapy resistance that affected prognosis.

Histone methylation is a reversible process subjected to demethylation or acetylation that regulates chromatin configuration and gene expression. Targeting histone methylation transferase EZH2 was a promising way to regulate histone modification H3K27me3 status and its gene-inhibiting function. According to current evidence, the effect of EZH2-H3K27me3 pathway to platinum resistance was heterogeneous in different cancer types: EZH2/H3K27me3 contributed to platinum resistance in ovarian cancer, cervical cancer, colorectal cancer, lung cancer and gastric cancer while sensitizing osteosarcoma and lymphoma to platinum treatment. Several possible mechanisms of EZH2 regulation of response to platinum chemotherapy were discovered so far. Enrichment of EZH2/H3K27me3 at promoter region of intrinsic apoptosis pathways (Caspase-9, p53, Bcl2 and Bax) and extrinsic apoptosis pathways (Fas) interfered the integrity of platinum induced apoptotic function and proper response to platinum treatment (Benard et al., 2014). In addition to apoptosis, EZH2/H3K27me3 pathway was also reported to negatively regulate autophagy pathway (Sun et al., 2016). Another possible mechanism was that inhibition of Dicer by EZH2 led to disordered miRNA function (Cai, Wang & Liu, 2016).

Canonical way of targeting EZH2 was primarily through molecular inhibitors such as DZNEP and GSK126. Mounting evidence suggested that EZH2 phosphorylation was involved in reprogram of H3K27Me3 profile and transcriptional activation. EZH2 phosphorylation at S21 by Akt1 impeded EZH2-H3 interaction leading to de-repression of silenced genes (Cha et al., 2005). EZH2 phosphorylation at T487 disrupted EZH2 binding with SUZ12 and EED, thereby inhibition of EZH2 methyltransferase activity (Wei et al., 2011). EZH2 phosphorylation at T311 by AMPK disrupted EZH2-SUZ12 interaction attenuated H3K27Me3. EZH2 phosphorylation at T372 by TNF-activated p38 α kinase promoted YY1-EZH2 interaction leading to the formation of repressive chromatin (Palacios et al., 2010). EZH2 phosphorylation at T350 contributed to the recruitment of EZH2 to EZH2-loci and maintenance of H3K27Me3 level (Chen et al., 2010). EZH2 phosphorylation at T345 by CDK1 increased EZH2-HOTAIR interaction (Kaneko et al., 2010). EZH2 phosphorylation at T345 and T487 by CDK1 promoted EZH2 ubiquitination and subsequent proteasomal degradation (Wu & Zhang, 2011).

To date, very few studies investigated the clinical implication of phosphorylated EZH2. Based on aforementioned evidences, the clinical effect of phosphorylated EZH2 was residue-specific. A recent study reported that EZH2 phosphorylation at T372 reduced ovarian cancer cell proliferation, migration and tumor formation (Wan et al., 2018). The levels of EZH2-T372 phosphorylation in primary ovarian tumor samples were significantly lower than that in normal ovarian surface epithelium (Ozes et al., 2018). Wan et al. reported that EZH2 phosphorylation at T311 by AMPK suppressed PRC2 activity and EZH2-pT311 correlated with better survival in ovarian and breast cancer patients (Wan et al., 2018). These results suggested that certain residue phosphorylation such as T311 and T372 antagonized PRC2 oncogenic effect by disrupting PRC2 complex and played a role as favorable prognostic factors. On the other hand, our results revealed that EZH2 phosphorylation at S21 associated with chemotherapy resistance and predicted poor PFS in ovarian cancer patients suggesting an oncogenic role of EZH2-pS21.

Ovarian cancer is a heterogeneous disease classified as several histology subtypes characterized by different molecular and biological features. In this study, we performed stratification analysis for chemotherapy response, OS and PFS within the serous ovarian cancer group in addition to general analysis. Low pS21EZH2 status remained predictive for better chemotherapy response and PFS. EZH2low/H3K27Me3low also showed predictive value regarding chemotherapy response although statistical analysis did not reach significant difference. While EZH2 and pS21EZH2 had inseparable correlation, pS21EZH2 was only positive in samples with positive EZH2. To investigate whether the predictive value of pS21EZH2 was EZH2 dependent, two different pathways were analyzed. EZH2/H3K27me3 combination provided better prognostic value than EZH2 alone while combination of pAkt1 and pS21-EZH2 was no gain. Considering that pAkt1 was a powerful multi-functional signal transducer regulating numerous pathways, the effect of other pAkt1 associated pathways might have some influence while evaluating pAkt1/pS21EZH2 pathway. Therefore, further cellular experiments were needed to elucidate the issue. Due to the limitation of number of cases included, the study only stratified the patients into serous/other groups failing to perform further analysis in other subtypes. Further studies with larger cohort might provide a more consolidate insight.

Conclusions

In conclusion, this study suggested that EZH2/H3K27Me3 and pEZH2 predicted chemotherapy response and progression-free survival in ovarian cancer.

Supplemental Information

File S1 Images with scores of the the percentage of positive tumor cells and stain intensity

Click here for additional data file.

File S2 Raw Data for basic characteristic infomation and statistical analyses

Click here for additional data file.

File S3 Negative control for IHC staining

Click here for additional data file.

Supplemental Information 4 MIAME checklist

Click here for additional data file.

We thank all recruited patients for providing tissue samples. We thank the staff and graduates from Department of Gynecology and Obstetrics, Union Hospital, Tongji Medical College, Huazhong University of Science and Technology for sample collection.

Additional Information and Declarations

Competing Interests

Author Contributions

Human Ethics

Data Availability

The authors declare there are no competing interests.

Si Sun conceived and designed the experiments, performed the experiments, analyzed the data, prepared figures and/or tables, authored or reviewed drafts of the paper, and approved the final draft.

Qiang Yang analyzed the data, authored or reviewed drafts of the paper, and approved the final draft.

E. Cai and Feiquan Ying performed the experiments, prepared figures and/or tables, and approved the final draft.

Bangxing Huang and Yiping Wen analyzed the data, prepared figures and/or tables, and approved the final draft.

Jing Cai and Ping Yang conceived and designed the experiments, authored or reviewed drafts of the paper, and approved the final draft.

The following information was supplied relating to ethical approvals (i.e., approving body and any reference numbers):

The Ethics Committee of Tongji Medical College, Huazhong University of Science and Technology approved the study (IOGR No: IORG0003571).

The following information was supplied regarding data availability:

The raw measurements are available in the Supplemental File.

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
