# Peer review of "EZH2/H3K27Me3 and phosphorylated EZH2 predict chemotherapy response and prognosis in ovarian cancer"

_PeerJ, doi:10.7717/peerj.9052_

## Round 0.1 · original submission · Major Revisions

There are some issues in the method for semi-quantitative IHC analysis.
1. Different background levels among samples are a major concern. Strong background in extracellular matrix (ECM) is particularly concerning. In all figures, p-AKT-high samples show signals in ECM, although p-AKT should be only present within cells.
2. The authors did not distinguish nuclear and cytoplasmic staining, and background and specific signal. Only nuclear staining should be considered as “specific” for H3K27Me3. However, the representative section show signal in cytoplasm as well as in ECM, raising an issue for the specificity.
I recommend the authors refine IHC analysis and measure: 1. cytoplasmic and nuclear signal levels in cancer cells, and 2. stromal and ECM signal intensity.
In such analyses, all images should be captured under identical conditions and processed by identical procedures.

Additionally, readers could more accurately evaluate the accuracy of the analysis if 1. a high magnification image and 2. the percentage of positive tumor cells and stain intensity scores for all IHC images used for the analyses were provided the in the supporting figure file.

·

Basic reporting

no comment.

Experimental design

no comment

Validity of the findings

no comment

Additional comments

This manuscript by Sun et al. investigated the correlation between the expression level of EZH2, pS21-EZH2, H3K27Me3 or pAkt and clinical behaviors of ovarian cancer. The data is clearly presented and the manuscript is well written. My primary concern is that different histologic subtypes of ovarian cancers arise from the different cells of origins and should be considered as distinct diseases. As the majority of the cases investigated in this study (53/65) were serous ovarian cancer, it is recommended that the authors should do analysis within the serous ovarian cancer and discuss the limitation of not a sufficient number of cases of other subtypes being included.

1. Fig2 and Fig1 used identical representative images for EZH2 and pEZH2. The images in Fig 2 provides little values and can be removed. Instead, the author should provide a correlation plot for EZH2 and chemo response. Given that both low pS21-EZH2 and low EZH2/lowH3K27Me3 correlated with better chemo-responses and survival, it will be interesting to analyze whether there is any correlation between pS21-EZH2 and EZH2 levels.
2. In Table 2, the combination of pAkt and pS21-EZH2 did not provide better value in predicting response to chemotherapy than pS21-EZH2 alone. The authors should comment on this in the discussion.
3. Line 169, the p-value is inconsistent with the value in Table 3.
4. Line 201. the word 'reprogram' is a verb and should be changed to a noun, reprogramming.

Reviewer 2 ·

Basic reporting

The manuscript was written adequately, however it requires editing to correct some language errors.

Experimental design

Some important details are missing in Materials and Methods (see General Comments to the Authors).

Validity of the findings

The Discussion contains some speculation, which would be hard to avoid considering the descriptive and associative nature of the study.

Additional comments

Sun and collaborators determined the associations between the EZH2 canonical and non-canonical signaling pathways with chemotherapy response and prognosis in ovarian cancer. For this, the authors measured levels of EZH2, H3K27me3, pAKT1, and pS21EZH2 by IHC in ovarian cancer tumors from 65 patients. The authors reported that EZH2/H3K27Me3 and pS21EZH2 IHC scores have predictive value for chemotherapy response and progression-free survival in ovarian cancer. Below are suggestions for improving the manuscript.

Comments
1. Indicate when the tumor samples were collected.
2. Lines 113-117. Provide the catalog numbers of the antibodies. Indicate what negative controls were included in the IHC runs.
3. Line 128. IHC scores (line 122), not protein expression, were used for determination of associations. IHC scores should be used throughout the manuscript when referring to protein levels in the tumors.
4. Lines 144-145. Indicate the p-values of the correlations to make designations of “closely” and “poorly” correlated less arbitrary.
5. Lines 154-155. When describing EZH2/H3K27Me3 or pAKT1/pS21EZH2 association with chemotherapy response, it is not clear if it is for each protein, either one, or both combined.
6. For consistency use either chemotherapy resistance or chemotherapy response throughout the manuscript.
7. Lines 197-198. Sentence is not clear. Please revise.
8. The Discussion should be expanded to include possible mechanisms of EZH2 regulation of response to platinum chemotherapy. Also, limitations of the study given the relatively small numbers of patients, and grouping of the samples only into serous and non-serous should be mentioned.
9. Figure 1. Pictures of low staining for p-EZH2, p-AKT1, and EZH2 show no staining. These pictures should be replaced with representative pictures that actually show some staining. Were the correlations actually calculated using percentages of positive cells as indicated on the figure legend? If percentages were in fact used justify why it was preferred over IHC scores.
10. Figure 2. Indicate how many patients were resistant or sensitive.
11. Table 1 and Table 2. Indicate the test corresponding to the P-values shown.

---

## Round 0.2 · Minor Revisions

The authors must address the editor's concern about the accuracy of immunohistochemical analyses before this manuscript is accepted.

·

Basic reporting

no more comments

Experimental design

no more comments

Validity of the findings

no more comments

Additional comments

The authors have addressed my concerns and made significant improvements to the manuscript.

Reviewer 2 ·

Basic reporting

The manuscript needs revision for English correctness.

Experimental design

No comment

Validity of the findings

Most comments of the first review were addressed satisfactorily by the authors. There is however a concern regarding the validation of immunohistochemistry for the different proteins measured here. This is important because the study is based mainly on the accuracy of these determinations. The use of PBS as a negative control to demonstrate antibody specificity is not appropriate; acceptable substitutes of primary antibody that can be used for this purpose are IgG or neutralization of primary antibody with immunizing peptide. The authors should run these tests for all antibodies used and include the results as supplementary information.

Additional comments

No additional comments

---

## Round 0.3 · Minor Revisions

Please respond to the editor's comments on the original version. I raised several critical issues regarding the validation of IHC. None of issues are addressed in two revisions.

---

## Round 0.4 · accepted · Accept

Thank you for your prompt reply. The issues are appropriately addressed. Congratulations!